# VISIOCITY: A New Benchmarking Dataset and Evaluation Framework Towards Realistic Video Summarization

**Vishal Kaushal**
Department of Computer Science and Engineering
Indian Institute of Technology Bombay, India
vkaushal@cse.iitb.ac.in

**Suraj Kothawade**
Department of Computer Engineering
University of Texas at Dallas, USA
suraj.kothawade@utdallas.edu

**Rishabh Iyer**
Department of Computer Science
University of Texas at Dallas, USA
rishabh.iyer@utdallas.edu

**Ganesh Ramakrishnan**
Department of Computer Science and Engineering
Indian Institute of Technology Bombay, India
ganesh@cse.iitb.ac.in

## Abstract

Automatic video summarization has attracted a lot of interest, but is still an unsolved problem due to several challenges. The currently available datasets either have very short videos or have a few long videos of only a particular type. We introduce a new benchmarking video dataset called VISIOCITY (VIdeo SummarIzatiOn based on Continuity, Intent and DiversiTY) which consists of longer videos across six different domains with dense concept annotations capable of supporting different flavors of video summarization and other vision problems. Secondly, supervised video summarization techniques require many human reference summaries as ground truth. Acquiring them is not easy, especially for long videos. We propose a strategy to automatically generate multiple reference summaries using the annotations present in VISIOCITY and show that these are at par with the human summaries. The annotations thus serve as *indirect* ground truth. Thirdly, due to the highly subjective nature of the task, different *ideal* reference summaries of long videos can be quite different from each other. Due to this, the current practice of evaluating a summary vis-a-vis a limited set of human summaries and over-dependence on a single measure has its shortcomings. Our proposed evaluation framework overcomes these and offers a better quantitative assessment of a summary's quality. Finally, based on the above observations we present insights into how a mixture model can be easily enhanced to yield better summaries and demonstrate the effectiveness of our recipe in doing so as compared to some of the representative state-of-the-art techniques when tested on VISIOCITY. We make VISIOCITY publicly available via our website[1].

## 1 Introduction and Motivation

Videos have become an indispensable medium for capturing and conveying information in many sectors like entertainment (TV shows, movies, etc.), sports, personal events (birthday, wedding etc.), education (HOWTOs, tech talks etc.), to name a few. However, the unprecedented rise in the amount of video data has also made it difficult to consume them. Most of this data comes with a lot of redundancy, partly because of the inherent nature of videos (as a set of *many* images) and partly due to the 'capture-now-process-later' mentality. This has given rise to the need for automatic

---

[1]https://visiocity.github.io/

Submitted to the 35th Conference on Neural Information Processing Systems (NeurIPS 2021) Track on Datasets and Benchmarks. Do not distribute.

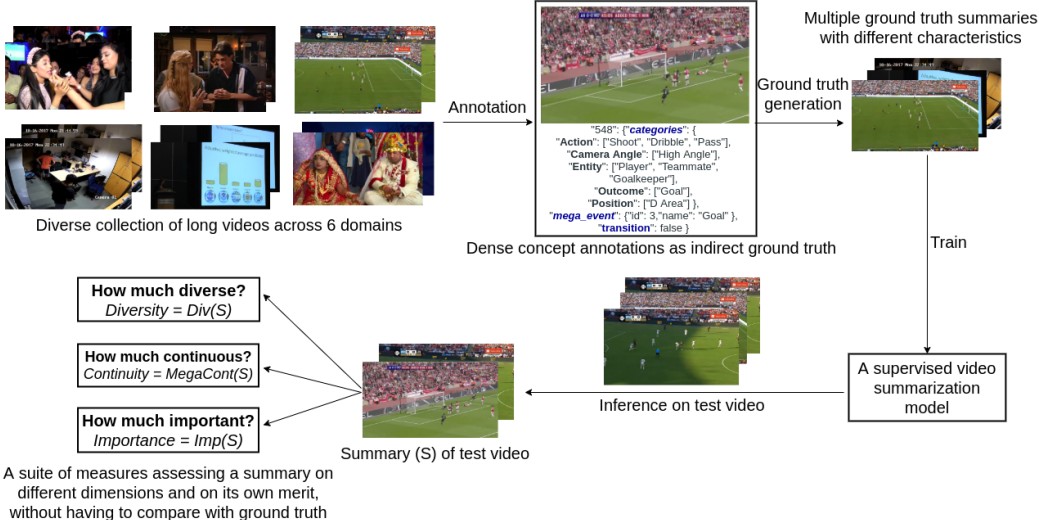

Figure 1: VISIOCITY at a glance

video summarization techniques which aim at producing much shorter videos without significantly compromising on the key information contained in them. For example, producing the highlights from a soccer video. A video summarization technique aims to select important, diverse (non-redundant) and representative elements (frames or shots) from a video to produce its summary. When the selections are frames, it is called *static video summarization* and when the selections are shots, it is called *dynamic video summarization*. In this work we focus on dynamic video summarization.

Though there has been a lot of work pushing the state-of-the-art for newer algorithms and model architectures [12, 4, 41, 40, 43, 10] and datasets [9, 27, 32], the literature also talks of a few fundamental challenges in automatic video summarization that need to be addressed before we have a more realistic video summarization that works in practice. **In this work, we introduce VISIOCITY as a step towards addressing the following challenges**:

**Lack of a challenging dataset**: Almost all recent techniques [24, 1, 12] have reported their results on TVSum [32] and SumMe [9] which have emerged as benchmarking datasets of sorts. However, since the average video length in these datasets is of the order of *only* 1-5 minutes, they are far from being effective in real-world settings. While there have been several attempts at creating better datasets for video summarization (Sec. 2), they either a) have very short videos, or b) have very few long videos of a particular type. **We introduce VISIOCITY which is a diverse collection of 67 long videos spanning across six different domains (Sec. 3)**. Since the videos span across different well-defined domains, VISIOCITY is also suitable for more in-depth domain specific studies on video summarization [34, 27, 40]. Secondly, different flavors of video summarization like query-focused video summarization [38, 31], are often treated differently and require different datasets. VISIOCITY provides dense concept annotations for each shot (Sec. 3). The concepts are carefully selected list of verbs and nouns based on the video domain (see Fig. 1 for example). In addition, there are higher-level annotations (which we call *mega-events*) that identify consecutive shots as events. Due to its rich annotations VISIOCITY can lend itself well to other flavors of video summarization and also other computer vision video analysis tasks like action recognition [37], event localization [6, 29, 7, 36], etc. We discuss other advantages of such annotations in Sec. 3. A large dataset with a lot of different types of full-length videos with rich annotations to be able to support different techniques was one of the recommendations in [34], is still not a reality, and is clearly a need of the hour [12]. VISIOCITY addresses this need.

**Challenges in evaluation**: The current practice is to use *reference based evaluation* [24] where a candidate summary is evaluated by comparing it against human summaries. However, video summaries are highly context dependent (that is, depend on the purpose behind getting a video summary), subjective (that is, even for the same purpose, preferences of two persons don't match) and depend on high-level semantics of the video (that is, two visually different scenes could capture the same semantics or visually similar looking scenes could capture different semantics). Hence, there is no single 'right answer' for a video and thus human summaries could be quite different in their

selections [13, 24], all the more so for long videos. Even if average or max is used to accommodate multiple human summaries [32, 10], a good candidate may get a low score just because it was not fortunate to have a matching human summary. Secondly, a typical measure used is F1 score defined as harmonic mean of precision (ratio of temporal overlap between candidate and reference summary to duration of summary) and recall (ratio of temporal overlap between candidate and reference summary to video duration) [42, 4, 12, 41, 40, 4, 12]. This has a couple of problems - a) due to the segmentation used as a post processing step in typical video summarization pipeline, even random summaries can get good F1 scores [24]; b) there are several desirable characteristics of a summary like *diversity* and *continuity* (Sec. 4) and F1 is not designed to measure them. For example, a summary should be diverse. That is, to be able to convey maximum information within a given budget, a good summary should prefer more diverse elements and minimize redundancy. Similarly, a summary should be as continuous as possible. A summary with more number of consecutive shots is more continuous (and hence pleasurable to watch). Two summaries may have same F1 score, and yet one may be more continuous than the other. To alleviate all these problems, in this work **we propose an evaluation framework (Sec. 4) which** a) **avoids over-dependence on one measure** by proposing a suite of measures to assess a summary on different dimensions; and b) **assesses a summary on its own merit** using the rich annotations in VISIOCITY instead of comparing it with one or more reference summaries.

**Difficulty in acquiring reference ground truth summaries for supervised learning**: Supervised techniques tend to work better than unsupervised techniques because of learning directly from human summaries [12, 41]. In a race to achieve better performance, most state-of-the-art techniques are based on deep architectures and are thus data hungry. Thus, more the number of human summaries, better is the learning. Unfortunately, for long videos getting human summaries is very time consuming. It becomes increasingly expensive and, beyond a point, infeasible to get these reference summaries from humans. Also, this is not scalable to experiments where reference summaries of different lengths are desired [10]. In this work **we propose a strategy based on the proposed measures to automatically generate ground truth reference summaries (Sec. 5) which can be used to train a model**.

We summarize the above aspects of VISIOCITY in Fig. 1. Using the above insights and leveraging VISIOCITY, as another contribution, **we demonstrate that better results can be achieved when a supervised model learns from individual diverse ground truth summaries** (instead of the typical practice of combining them into one *oracle* summary [41, 4, 12]) **and using a combination of losses, each measuring deviation from different desired characteristics of summaries (Sec. 6).**

## 2 Related Work

**Datasets:** One of the prominent problems in video summarization literature has been a lack of a standardized benchmarking dataset. Because of this, in proposing new techniques of summarization, researchers often created new datasets. Table 1 compares VISIOCITY with other existing datasets for video summarization. The 6 genres of VSumm(YouTube) [2] are cartoons, news, sports, commercials, tv-shows and home videos and the 5 genres of VSumm(OVP) [2] are documentary, educational, ephemeral, historical, lecture. The UGSum52 [19] videos are distributed across holiday, events and sports. Textual descriptions for each 5 sec snippet of UTE [18] videos are provided by [39]. We note the following - a) though the number of categories in TVSum [32] and MED Summaries [27] appear to be large, the notion of categories there is of events, like 'making a sandwich' or 'attempting bike tricks', quite different from the notion of *domains* in VISIOCITY with an intent of studying the characteristics of summaries of different types of videos like sports or TV Shows; b) LOL [5] dataset contains online eSports videos from the League of Legends. While this dataset is significantly larger compared to the other datasets, it is limited only to a single domain, i.e. eSports; c) Due to its advantages, indirect ground truth as annotations has been recommended by [34]. While SumMe, VSumm(OVP), VSumm(YouTube), Tour20, LOL and UGSum52 provide direct ground truth in the form of human summaries, MEDSummaries and TVSum provide indirect ground truth in form of scores. VISIOCITY on the other hand provides indirect ground truth as dense concept annotations for every shot which has its unique advantages (Sec. 3). For the purpose of query-focused summarization, [30] have extended the UTE dataset [18] to provide concept annotations for each 5 sec snippet but the dataset is still limited to only egocentric videos and does not support any concept hierarchy in the annotations. To the best of our knowledge, VISIOCITY is one of its kind large dataset with many long videos spanning across multiple domains and annotated with dense concept annotations for each shot.

| Name | # Videos | Avg Duration | Types of Videos | Type of Annotation |
|---|---|---|---|---|
| MEDSummaries [27] | 160 | 1-5m | 15 event categories | Segments and their importance scores |
| TVSum [32] | 50 | 4m | 10 event categories | Importance scores of every 2s snippets |
| SumMe [9] | 25 | 2m | Misc. | 15-18 summaries/video |
| VSumm(OVP) [2] | 50 | 1-4m | 5 genres | 5 summaries/video |
| VSumm(YouTube) [2] | 50 | 1-10m | 6 genres | 5 summaries/video |
| UTE [18] | 4 | 254m | Egocentric | Text [39] or concepts [30] for every 5s snippets |
| Tour20 [25] | 140 | 3m | Tourist places | 3 summaries/video |
| TV Episodes [39] | 4 | 45m | TV shows | Text for every 10s snippets |
| LOL [5] | 321 | 30-50m | eSports | Summaries |
| UGSum52 [19] | 52 | 4m | 3 categories of user videos | 25 summaries per video |
| VISIOCITY | 67 | 55m | 6 domains | Concepts for every shot |

Table 1: VISIOCITY has many long videos spanning across multiple domains and annotated with dense concept annotations for each shot

**Techniques for Automatic Video Summarization**: A lot of past work exists for automatic video summarization for example, using submodular functions [41, 10, 14, 10, 15], LSTMs [41], reinforcement learning [43] and attention models [12, 4]. vsLSTM [41] is a supervised technique that uses BiLSTM to learn the variable length context in predicting important scores. It learns from a combined ground truth in terms of aggregated scores. VASNet [4] is a supervised technique based on a simple attention based network without computationally intensive LSTMs and BiLSTMs. It learns from a combined ground truth in terms of aggregated scores and outputs a predicted score for each frame in the video. DR-DSN [43] is an unsupervised deep-reinforcement learning based model which learns from a combined diversity and representativeness reward on scores predicted by a BiLSTM decoder. It outputs predicted score for every frame of a video. We demonstrate the effectiveness of our recipe in improving a mixture model to achieve better results than vsLSTM, VASNet and DR-DSN when tested on VISIOCITY.

**Evaluation**: Early approaches [21, 22] involved user studies but suffered the obvious demerit of cost and reproducibility. With a move to automatic evaluation, every new technique of video summarization came with its own evaluation criteria making it difficult to compare results different techniques. VIPER [3] addressed the problem by defining a specific ground truth format which makes it easy to evaluate a candidate summary, and SUPERSEIV [11] which is an unsupervised technique to evaluate video summarization algorithms that perform frame ranking. VERT [20] on the other hand was inspired by BLEU in machine translation and ROUGE in text summarization. Other techniques include pixel-level distance between keyframes [16], objects of interest as an indicator of similarity [18] and precision-recall scores over key-frames selected by human annotators [8]. More recently, computing overlap between groundtruth and generated summaries reported by F-measure has become the standard framework for video summary evaluation [42, 4, 12, 41, 40, 4, 12]. Yet others prefer to evaluate a summary in the text domain as text is better at capturing higher level semantics [39, 26]. This also forms the motivation behind our proposed evaluation measures. However, our measures are different in the sense that a summary is not converted to text domain before evaluating. Rather, how important its selections are, or how diverse its selections are, is computed from the rich textual annotations in VISIOCITY. This is similar in spirit to [30], but there it was done only for egocentric videos.

## 3 VISIOCITY Dataset

**Videos:** VISIOCITY is a diverse collection of 67 long videos spanning across six different domains: TV shows (*Friends*) , sports (soccer), surveillance, education (tech-talks), birthday videos and wedding videos. Summary statistics for videos in VISIOCITY are presented in Table 2. Publicly available soccer, tech-talk, birthday and wedding videos with Creative Commons CC-BY (v3.0) license were downloaded from YouTube. Only high resolution videos which were long enough were retained. Soccer videos typically have well-defined events of interest like goals or penalty kicks and are very similar to each other in terms of the visual features. VISIOCITY includes diverse

soccer videos covering different events including score changing events, non-score changing events, pre & post celebrations and even matches where no goals were scored. Under TV shows domain, VISIOCITY contains purchased videos from a popular TV series *Friends*. They are typically more aesthetic in nature and professionally shot and edited. Birthday and wedding videos on the other hand are typically long and unedited. VISIOCITY contains diverse birthday videos spanning birthdays of public figures (3), boy (2), girl (2) and lady (2). Wedding videos are from diverse cultural backgrounds - Bengali (1), North Indian (5), South Indian (2) and Christian (2). Under surveillance domain, VISIOCITY covers 2 outdoor videos and diverse indoor videos - classroom (2), office (4) and lobby (4). The videos were recorded by us at our premises using our own surveillance cameras with the permission of the subjects. These videos are in general very long and are mostly from static continuously recording cameras. Under educational domain, VISIOCITY has diverse tech-talk videos with different views like both speaker and presentation visible, either speaker or presentation visible, talk in auditorium, speaker in frame inset, etc. All videos were processed to remove the audio. We used Kernel Temporal Segmentation (KTS) [27] to mark the shots in the video. For surveillance videos, which are with static cameras, we use fixed 2 seconds snippets as shots. The videos and the shots information are accessible from the project website at https://visiocity.github.io/

**Annotations:** VISIOCITY provides dense concept annotations for each shot in the videos instead of the summaries themselves. Concepts are a carefully selected list of verbs and nouns based on the type of the video and are given importance ratings based on the knowledge of the particular domain. The concepts are organized in categories instead of a long flat list. Example categories include 'actor', 'entity', 'action', 'scene', 'number-of-people', etc. (see for example, Fig. 1). Categories provide a natural structuring to make the annotation process easier and

| Domain | # Videos | Duration (min,max,avg) in minutes | Total Duration |
|---|---|---|---|
| Soccer | 12 | (37,122,**64**) | 12.77 h |
| Friends | 12 | (22,26,**24**) | 4.74 h |
| Surveillance | 12 | (22,63,**53**) | 10.55 h |
| Educational | 11 | (15,122,**67**) | 12.22 h |
| Birthday | 10 | (20,46,**30**) | 4.87 h |
| Wedding | 10 | (40,68,**55**) | 9.15 h |
| All | 67 | (15,122,**49**) | 54.31 h |

Table 2: Key Statistics of VISIOCITY.

also provide support for at least one level hierarchy of concepts for query-focused summarization. In addition to concepts, we ask annotators to group those consecutive shots as *mega-events* which together constitute a cohesive event. For example, a few shots preceding a goal in a soccer video, the goal shot and a few shots after the goal shot together would constitute a 'mega-event'. The prefix 'mega' refers to the fact that it is not an annotation of a shot per se but is a higher level annotation corresponding to a group of shots. A model trained to learn importance scores (only) would do well to pick up the 'goal' shot. However, such a summary will not be very pleasing to watch because what is required in a summary in this case is not just the ball entering the goal post, but the build up to this event and probably a few shots as a followup. Thus, this notion of mega events helps us to model the notion of continuity.

**Annotation Protocol and Quality of Annotations:** A group of 13 professional annotators were tasked to annotate videos (without the audio) by marking all applicable keywords on a shot through a python GUI application developed by us for this task. It allows an annotator to go over the video shot by shot and select the applicable keywords using a simple and intuitive GUI. It provides convenience features like copying the annotation from a previous shot, which comes in handy where there are a lot of consecutive identical shots, for example in surveillance videos. The annotation guidelines and protocols were made as objective as possible, the annotators were trained through sample annotation tasks, and the annotation round was followed by two verification rounds where both 'precision' (whether the marked annotations were correct) and 'recall' (whether all events of interest and continuity information in the video has been captured in the annotations) were manually verified by another set of annotators.

**Advantages of concept annotations in VISIOCITY:** This kind of annotation allows for generating multiple reference summaries of different lengths with different desired characteristics and is easy to scale (Sec. 5). For long videos, acquiring such an indirect ground truth is more objective and easier than asking the annotators to produce reference ground truth summaries. While past work has made use of other forms of indirect ground truth like asking annotators to give a score or a rating to each shot [27, 32], using textual concept annotations in particular offers several advantages. First, especially for long videos, it is easier and more accurate for annotators to mark all keywords applicable to a shot than for them to tax their brain and give a rating (especially when it is quite subjective and requires going back and forth over the video for considering what is *more important*

or *less important*). Second, when annotators are asked to provide ratings, they often suffer from chronological bias [32]. [32] addresses this for 4 min. videos by showing the snippets to the annotators in random order but it doesn't work for long videos because an annotator cannot remember all of these to be able to decide the relative importance of each. Third, the semantic content of a shot is better captured through text [39, 26]. Two shots may look visually different but could be semantically same and vice versa. Text captures the right level of semantics desired by video summarization. Also, when two shots have the same rating, it is not clear if they are semantically same, or they are semantically different but equally important. Textual annotations bring out such similarities and dissimilarities more effectively. Fourth, as already noted, textual annotations make it easy to adapt VISIOCITY to a wide variety of problems.

# 4 Proposed Evaluation Framework

Video summarization literature talks about certain desirable good characteristics of a video summary [10, 16, 18, 22, 40, 43]. For example, a good video summary is supposed to be diverse (non-redundant), continuous or visually pleasing (without abrupt shot transitions), representative of the original video and contain impor-

| Measure | Expression |
|---|---|
| DiversitySim (DS) | $\min_{i,j \in X} d_{ij}$ |
| Diversity(Time/Concept) (DT/DC) | $\sum_{i=1}^{|C|} \max_{j \in X \cap C_i} r_j$ |
| Mega Event Continuity (MC) | $\sum_{i=1}^{E} r^{mega}(M_i)|X \cap M_i|^2$ |
| Importance (IMP) | $\sum_{s \in X \cap A \setminus M} r(s)$ |

Table 3: Proposed measures in VISIOCITY.

tant or interesting shots from the video. In what follows, we propose the measures to assess the candidate summaries on these characteristics and summarize them in Table 3.

**Diversity:** Let $V$ be a video (a set of shots) and $X \subset V$ be a summary. $X$ is diverse if it contains segments quite *different* from one another. When the similarity is measured in terms of the content alone, we call it $Div_{sim}(X)$ and measure it as $Div_{sim}(X) = \min_{i,j \in X} d_{ij}$ where $d_{ij}$ is IOU based distance measure between shots $i$ and $j$ represented by binary concept vectors based on their concept annotations. This is a typical notion of diversity. For example, in the summary of a *Friends* video, given a fixed budget, one may want to see different kinds of shots instead of too many similar looking shots. However, in some other domain, say surveillance, consider a video showing a person entering her office at three different times of the day. Though all three look similar (and will have identical concept annotations as well), all could be desired in the summary for the summary of surveillance to be effective. Thus, one may want a summary which doesn't have too many similar consecutive shots but does have similar shots that are separated in time. We call this flavor of diversity $Div_{time}$ and measure it as $Div_{time}(X) = \sum_{i=1}^{|C|} \max_{j \in X \cap C_i} r_j$ where $C$ are the clusters, which are defined over time. That is, all consecutive shots with same set of concept annotations form a cluster. $r_j$ is the importance rating of a shot $j$. On similar lines, this notion of diversity can be extended to the concept covered by the shots. One may not want too many shots covering the same concept and would rather want a few shots from all concepts. We define this notion of diversity as $Div_{concept}$ and measure it as $Div(X) = \sum_{i=1}^{|C|} \max_{j \in X \cap C_i} r_j$ where the clusters are now defined over concepts. That is, all shots which have been marked with a particular concept belong to a cluster for that concept. In this case there are as many clusters as the total number of concepts. When optimized, this function leads to the selection of the best shot from each cluster. However, this can be easily extended to select a finite number of shots from each cluster instead of the best one.

**MegaEventContinuity:** element of continuity makes a summary pleasurable to watch. Since only a small number of shots are to be included in a summary, some discontinuity in the summary is expected. However, the less the discontinuity at a semantic level, the more pleasing is the summary to watch. There is a thin line between modelling redundancy and continuity. Some shots might be redundant but are important to include in the summary from a continuity perspective. To model the continuity, VISIOCITY has the notion of mega-events as defined earlier. To ensure no redundancy *within* a mega event, the mega-event annotations are as tight as possible, meaning they contain bare minimum shots just enough to indicate the event. A non-mega event shot is continuous enough to exist in the summary on its own and a mega event shot needs other adjacent shots to be included in the summary for semantic continuity. We measure mega-event continuity as follows: $MegaCont(X) = \sum_{i=1}^{E} r^{mega}(M_i)|X \cap M_i|^2$ where, $E$ is the number of mega events in the video annotation, $r^{mega}(M_i)$ is the rating of the mega event $M_i$ and is equal to $\max_{\forall s \in M_i} r(s)$, $A$ is the annotation of video $V$, that is, a set of shots such that each shot $s$ has a set of keywords $K^s$

and information about mega event, $M$ is a set of all mega events such that each mega event $M_i$ ($i \in 1, 2, \cdots E$) is a set of shots that constitute the mega event $M_i$

**Importance** - This is the most obvious characteristic of a good summary. For some domains like sports, there is a distinct importance of some shots over other shots (for e.g. score changing events). This however is not applicable for some other domains like tech talks where there are few or no distinctly important events. With respect to the annotations available in VISIOCITY, the importance of a shot is defined by the ratings of the keywords of that shot. These ratings come from a mapping function which maps keywords to ratings for a domain. The ratings are defined from 0 to 10 with 10 rated keyword being the most important and 0 indicated an undesirable shot. We assign ratings to keywords based on their importance to the domain and average frequency of occurrence. Given the ratings of each keyword, rating of a shot is defined as $r_s = 0$ if $\exists i : r_{K_i^s} = 0$, and $r_s = \max_i r_{K_i^s}$ otherwise. Here $K^s$ is the set of keywords of a shot $s$ and $r_{K_i^s}$ is the rating of a particular keyword $K_i^s$. Thus, importance function can be defined as: $\text{Imp}(X) = \sum_{s \in X \cap A \setminus M} r(s)$. Note that when both importance and mega-event-continuity is measured, we define the importance only on the shots which are non mega-events since the mega-event-continuity term above already takes care of the importance of the mega-event shots.

As discussed earlier, since there are multiple "right" answers with varying characteristics, we hypothesize that these are orthogonal characteristics and vary across different human (good) summaries. For example, one human summary could contain more important but less diverse segments while another human summary could contain more diverse and less important segments depending on the intent behind the summarization or user subjectivity. Also, in assessing summaries, one measure could be more relevant than another depending on the type of the video. For example, in sports videos because of well-defined events of interest, importance is more relevant in evaluating a summary. We empirically verify our hypotheses in Sec. 7. Hence, we propose that a true and wholesome assessment of a candidate summary can only be done when this suite of measures (including the existing measures like F score) are used instead of depending on only one measure. Results and observations from our extensive experiments corroborate this fact.

# 5 Ground Truth Summaries for Supervised Learning

In practice, it is difficult to acquire many human summaries with diverse characteristics, especially for long videos. We propose a strategy to automatically generate the reference ground truth summaries of desired lengths using the annotations present in VISIOCITY. Specifically, we use the above proposed evaluation measures as scoring functions and maximize them to get the desired ground truth summaries. We note that maximizing a particular scoring function would yield a summary rich in that particular characteristic, but it may fall-short on other characteristics. For example, a summary maximizing importance alone will select the goal shots from a soccer video, but some shots preceding the goal and following the goal will not be in the summary and the summary will not be visually pleasing (example illustration at https://visiocity.github.io/). Hence, a weighted mixture of such measures is used as a composite scoring function. Mathematically, given $X$, a set of shots of a video $V$, let $score(X)$ be defined as: $score(X, \Lambda) = \lambda_1 MegaCont(X) + \lambda_2 Imp(X) + \lambda_3 Div_{sim}(X) + \lambda_4 Div_{time}(X) + \lambda_5 Div_{concept}(X)$. This composite scoring function parameterized on $\lambda$'s takes an annotated video (*keywords* and *mega-events* defined over shots) and is approximately maximized via a greedy algorithm [23] to arrive at the ground truth summary. Different configuration of $\lambda$s generates different summaries. We use the notion of *Pareto optimality* to arrive at optimal configurations to be used. Pareto optimality is a situation that cannot be modified so as to make any one individual or preference criterion better off without making at least one individual or preference criterion worse off. Beginning with a random element (a possible configuration of the $\lambda$s) in the pareto-optimal set, we iterate over remaining elements to decide whether a new element should be added or old should be removed, or a new element should be discarded. This is decided on the basis of the performance of that element (configuration) on various measures. A configuration is better than another only when it is better on all measures, otherwise it is not. We use the summaries generated by the pareto-optimal configurations as ground truth summaries. We verify experimentally that the automatic ground truth summaries so generated are at par with the human summaries both qualitatively and quantitatively (Sec. 7). We use them in training the models tested on VISIOCITY.

## 6 Towards A New State of the Art

We apply two ideas to propose a recipe for a new state-of-the-art model. Firstly, most supervised learning approaches combine several ground truth summaries into one *oracle* summary [41, 4, 12, 40]. This suppresses the separate flavors captured by each of them. This was also noted by [1, 43] where they argue that supervised learning approaches, which rely on the use of a combined ground-truth summary, cannot fully explore the learning potential of such architectures. The necessity to deal with different kind of summaries in different ways was also observed by [34]. In fact, [1, 43] use this argument to advocate against the use of supervised approaches. Secondly, a model would do well if it receives feedback from a combination of losses, each measuring the deviation from different desired characteristics. We employ the strategy of large-margin learning of mixtures as proposed by [35, 10] and apply these ideas therein. Specifically, given a video $V$ as a set of shots $Y_v$, the problem reduces to picking $y \subset Y_v$ which maximizes the weighted mixture such that $|y| \leq k$, $k$ being the budget. That is, $y^* = \operatorname{argmax}_{y \subseteq Y_v, |y| \leq k} o(x_v, y)$, where, $y^*$ is the predicted summary, $x_v$ the feature representation of the video shots and $o(x_v, y) = w^T f(x_v, y)$ is the weighted mixture of components. We use a submodular facility-location term and modular importance terms as components of the mixture. The facility location function is defined as $f_{fl}(X) = \sum_{v \in V} \max_{x \in X} sim(v, x)$ where $v$ is a shot from the ground set $V$ and $sim(v, x)$ measures the cosine-similarity between shot $v$ and shot $x$ represented as concept-vectors. Facility-location thus models representativeness. During training and inference, these concept vectors are computed based on the detections from a YOLOv3 object detection model [28] pre-trained on the open images dataset [17]. The importance scores of shots are taken from the VASNet model [4] and the vsLSTM model [41] trained on VISIOCITY. The weights of the model are learnt using the large margin framework as described in [10] using many automatic ground truth summaries and a margin loss which combines the feedback from the proposed evaluation measures. Specifically, given $N$ pairs of a video and an automatic reference summary $(V, y_{gt})$, we learn the weight vector $w$ by optimizing the following large-margin formulation [33]: $\min_{w \geq 0} \frac{1}{N} \sum_{n=1}^{N} L_n(w) + \frac{\lambda}{2} ||w||^2$, where $L_n(w)$ is the generalized hinge loss of training example $n$ and $w$ is the weight vector. That is, $L_n(w) = \max_{y \subseteq Y_v^n} (w^T f(x_v^n, y) + l_n(y)) - w^T f(x_v^n, y_{gt}^n)$. For training example $n$, the margin loss we choose is a linear combination of the normalized losses reported by our proposed measures (Tab. 3). We call our proposed method VISIOCITY-SUM. We show that a simple model like this out-performs the current techniques (state of the art on TVSum and SumMe) on VISIOCITY dataset.

## 7 Experiments and Results

We asked a set of 11 users (different from the annotators) to create human summaries for two randomly sampled videos of each domain. The users were asked to look at the video without the audio and mark segments they feel should be included in the summary such that the length of the summary remains between 1% to 5% of the original video. The procedure followed was similar to that of SumMe [9]. F1 score of any

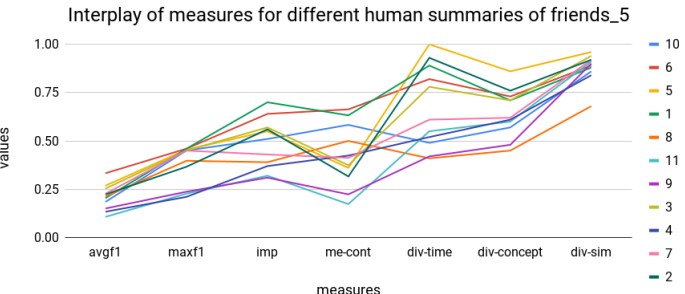

Figure 2: Different human summaries of same video perform differently on different measures.

summary was computed with respect to the human ground truth summaries following [41]. We report both avg F1 and max F1. To calculate F1 scores of a human summary with respect to human summaries, we compute max and avg in a leave-one-out fashion. In all tables, AF1 refers to Avg F1 score, MF1 refers to Max F1 score (nearest neighbor score), IMP, MC, DT, DC and DSi refer to the importance score, mega-event continuity score, diversity-time score, diversity-concept score and diversity-similarity score respectively, as calculated by the proposed measures(Sec. 4). All figures are in percentages. All experiments were run on a NVIDIA RTX 2080Ti GPU.

## 7.1 Different human summaries have different characteristics

We assess these human summaries qualitatively and quantitatively using the proposed set of evaluation measures. The human summaries were found to be consistent with each other in as much as there are important scenes in the video, for example, goals in Soccer videos (illustrative example on project website). In the absence of such clear interesting events, the human summaries exhibit more inconsistency with each other. A representative plot (for the scores of 11 human summaries of "friends_5" video is presented in Figure 2). As expected, we see that different human summaries of same video perform differently on different measures.

## 7.2 Automatically generated reference summaries are at par with human summaries

We compare automatically generated reference summaries with human summaries across all domains and present the results in Table 4. We see that the automatically generated summaries are much better than uniform summaries and random summaries and are at par with the human summaries. This is also confirmed in Figure 3 where we report detailed results on all measures for soccer videos. Again we see that

| Domain | Fri | Soc | Wed | Surv | TechT | Bday |
|--------|-----|-----|-----|------|-------|------|
| Human | 24 | 30 | 21 | 35 | 20 | 21 |
| Uniform | 5 | 6 | 5 | 6 | 7 | 6 |
| Random | 6 | 5 | 5 | 6 | 6 | 6 |
| *Auto* | 25 | 27 | 14 | 31 | 25 | 17 |

Table 4: Performance (AF1) of human summaries and automatically generated ground-truth summaries on videos across all the domains.

the proposed measures get good values for automatic ground truth summaries and human summaries as compared to random. Further, the automatic ground truth summaries have the highest importance, continuity and diversity scores. This is not surprising as they are obtained at the first place by optimizing a combination of these criteria.

We also compare the human and automatic summaries qualitatively. We present some results in the project page. We see a considerable similarity in selections, though a perfect match of selections is neither possible nor expected, in keeping with the spirit of multiple correct answers. Some human summary videos and automatic ground truth summary videos are also reported at the project page. We see that a) it is very hard to distinguish the automatic summaries from human summaries and b) they form very good visual summaries in themselves.

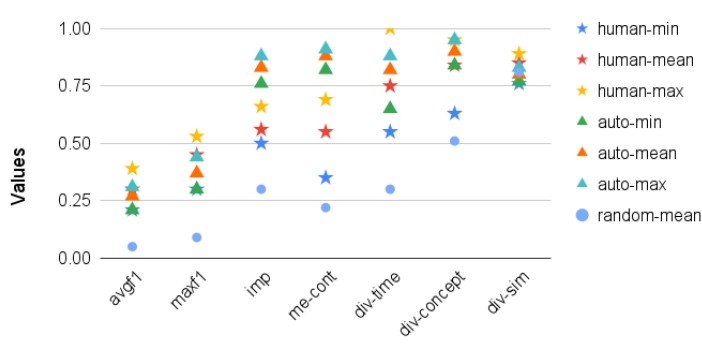

Figure 3: Performance of different types of summaries of Soccer videos.

## 7.3 VISIOCITY Benchmark: Performance of different models on VISIOCITY

| Method | SumMe | TVSum |
|--------|-------|-------|
| vsLSTM | 41.6 | 57.9 |
| VASNET | 51.09 | 62.37 |
| DR-DSN | 43.9 | 59.8 |

Table 5: F1 Scores as reported in respective papers

We test the performance of three different representative state-of-the-art techniques vsLSTM, VASNET and DR-DSN on the VISIOCITY benchmark. Along with the proposed measures, we report their avg and max F1 scores which we compute against the automatically generated summaries as a proxy for human summaries. We generate 100 automatic ground truth summaries for each video such that their lengths are 1% to 5% of the video length. For every domain and for every model, we report these measures averaged across $k$ runs of leave-one-out cross validation, $k$ being the number of videos in that domain. We follow [41] to convert importance scores predicted by vsLSTM, VASNET and DR-DSN to generate a predicted summary of desired length (max 5% of original video). Our proposed model, VISIOCITY-SUM learns from multiple ground truth summaries using Nesterov's accelerated gradient descent and outputs a machine generated summary as a subset of shots for a test video. For brevity here we

report the numbers for soccer and friends videos and defer the rest to the Supplementary.We make the following observations: a) DR-DSN tries to generate a summary which is diverse. As we can see in the results, it almost always gets high score on the diversity term. Please note that the way we have defined these diversity measures, diversity-concept (DC) and diversity-time (DT) have an element of importance in them also. On the other hand, diversity-sim (DSi) is a pure diversity term where DR-DSN almost always excels. b) Due to this nature of DR-DSN, when it comes to videos where the interestingness stands out and importance clearly plays a more important role, DR-DSN doesn't perform well. In such scenarios, vsLSTM is seen to perform better, closely followed by VASNET. c) It is also interesting to note that while two techniques may yield similar scores on one measure, for example vsLSTM and VASNET for Soccer videos (Table 6), one of them, in this case vsLSTM, does better on mega-event continuity and produces a desirable characteristic in the summary. This further strengthens our claim of having a set of measures evaluating a technique or a summary rather than over dependence on one, which may not fully capture all desirable characteristics of good summaries. d) We also note that even though DR-DSN is an unsupervised technique, it is a state of the art technique when tested on tiny datasets like TVSum or SumMe, but when it comes to a large dataset like VISIOCITY, with more challenging videos, it doesn't do well, especially on those domains where there are clearly identifiable important events for example in Soccer (goal, save, penalty etc.) and Birthday videos (cake cutting, etc.). In such cases, models like vsLSTM and VASNET perform better as they are geared towards learning importance. In contrast, since the interestingness level in videos like Surveillance and Friends is more spread out, DR-DSN does relatively well even without any supervision. e) VISIOCITY-SUM does better than all techniques on account of learning from individual ground truth summaries and a combination of loss functions. We also report the performance of these techniques on TVSum and SumMe as published in the respective papers in Tab. 5. Though not directly comparable in our settings, we see that while they measured their success on SumMe and TVSum, their strengths and weaknesses are better highlighted when tested on VISIOCITY.

# 8 Conclusion

We presented VISIOCITY, a large benchmarking dataset and evaluation framework and demonstrated its effectiveness in real world setting. To the best of our knowledge, it is the first of its kind in the scale, diversity and rich concept annotations. We introduce a strategy to automatically create ground truth summaries typically needed by the supervised techniques. Motivated by the fact that different good summaries have different characteristics and are not necessarily better or worse than the other, we propose an evaluation framework

| Domain | Technique | AF1 | MF1 | IMP | MC | DT | DC | DSi |
|--------|-----------|-----|-----|-----|-----|-----|-----|-----|
| Soccer | Auto | 59.3 | 93.3 | 83.2 | 84.3 | 82.6 | 85.9 | 76.2 |
| | DR-DSN | 2.8 | 8.9 | 23.7 | 20.3 | 23.2 | 30.4 | **83.4** |
| | VASNET | 28.4 | 43.4 | **63** | 49.3 | **62.1** | **67.4** | 75.2 |
| | vsLSTM | **31.9** | **48.2** | 62.2 | 60.1 | 62 | 69.5 | 76.5 |
| | **Ours** | 32.6 | 50.3 | 64.2 | 62.6 | 63.4 | 72.2 | 78.7 |
| | Random | 3.4 | 9.3 | 25.7 | 18.5 | 25.5 | 39.2 | 80.5 |
| Friends | AUTO | 66.3 | 96.9 | 87.8 | 84.6 | 80.3 | 89.8 | 83.1 |
| | DR-DSN | 4.3 | 9.4 | 19.1 | 6.9 | **65.7** | 51.5 | **98.5** |
| | VASNET | **17** | **29.6** | **41** | **49** | 39.3 | **60.6** | 86.7 |
| | vsLSTM | 15.5 | 27.2 | 40.4 | 39.2 | 64.7 | 59 | 91.1 |
| | **Ours** | 17.4 | 31.2 | 42.5 | 40.5 | 50.2 | 64 | 90.3 |
| | Random | 7.7 | 17.9 | 31.5 | 19.8 | 34.8 | 45.2 | 85.9 |

Table 6: Comparison of different techniques on VISIOCITY for Soccer and Friends videos. Results for other domains are in the Supplementary.

better geared at modeling human judgment through a suite of measures than having to overly depend on one measure. Finally we report the strengths and weaknesses of some representative state of the art techniques when tested on this new benchmark and demonstrate the effectiveness of our simple extension to a mixture model making use of individual ground truth summaries and a combination of loss functions. We hope our attempt to address the multiple issues currently surrounding video summarization as highlighted in this work, will help the community advance the state of the art in video summarization. We make VISIOCITY available through the project page at https://visiocity.github.io/.

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
