# OpenReview forum: "VISIOCITY: A New Benchmarking Dataset and Evaluation Framework Towards Realistic Video Summarization"
_NeurIPS.cc/2021/Track/Datasets_and_Benchmarks/Round1 — Submitted to NeurIPS 2021 Datasets and Benchmarks Track (Round 1)_

### Official Review · Reviewer_iGgd · 2021-07-05
**VISIOCITY: A New Benchmarking Dataset and Evaluation Framework Towards Realistic Video Summarization**

**Rating:** 4
**Confidence:** 3
**Correctness:** Yes, the dataset appears to be reason…

**Strengths:**

The authors presented a large-scale dataset with dense annotations for video summarization task, in which the videos are very long comparing to the previous works.

**Weaknesses:**

The claim on ln 155 seems to be a bit contradictory to the claim made in the Abstract, where "indirect ground truth" is advocated. But starting in ln 155, the author claims that the "textual concept annotations" also have some advantages. It is a bit confusing which method the authors eventually went with.

I am not too familiar with the practices of video summary but I do recommend the authors adding a brief description of what the task really is. Is the task summarizing a video into text? or is it extracting the important frames out of a given video.

From ln 199 to ln 204, the description of the preference of diversity is confusing. Why does one not want two similar-looking snippets to appear in a summary if they are indeed more important for video summary when connected? and why does one want them to be separated across time?

The paper lacks the visualizations of the concepts, tasks, procedures that are introduced in the paper, which otherwise could significantly help with the understanding of the authors' intent.

**Additional Feedback:**

Please see above.

**Clarity:**

There are some sections that are contradictory, as described in the weakness section above. Also, the paper is missing any visualization that could help understand the paper.

**Documentation:**

In ln 186 to ln 192, the procedure for annotation is described. How were the precision and recall for the annotators determined? How were the ground truths annotated in the first place for analyzing the precision and recall? If there already existed some ground truths to determine these metrics, why is it necessary to hire annotators to re-annotate the data?

**Relation To Prior Work:**

Yes, the differences are highlighted.

**Summary And Contributions:**

The paper proposes a large-scale dataset for the video summarization task. The dataset mainly consists of very long videos with indirect ground truth annotations. The paper fills the gap where there exists no densely annotated long-range video dataset for video summarization

---

> ### Author Response · Authors · 2021-07-11
> **Our responses to questions/concerns raised (part 1)**
>
> Thank you for your valuable comments. We address your questions/concerns as follows and are incorporating the changes in the modified version and shall submit the same before the discussion phase ends.
>
> >> "The claim on ln 155 seems to be a bit contradictory to the claim made in the Abstract, where "indirect ground truth" is advocated. But starting in ln 155, the author claims that the "textual concept annotations" also have some advantages. It is a bit confusing which method the authors eventually went with."
>
> Looks like there is a misunderstanding here. Indirect ground truth could refer to anything different from the actual summaries themselves. For example, textual annotations, scores, ratings etc. The notion of “indirect ground truth” was one of the recommendations by [27], specifically in their Sec. 5.4.2. VISIOCITY uses textual annotations as indirect ground truth and in the paragraph starting on line no. 155, we are enumerating the advantages of doing so as against using other forms of indirect annotations. We are making it more clear in the modified version.
>
> >> "I am not too familiar with the practices of video summary but I do recommend the authors adding a brief description of what the task really is. Is the task summarizing a video into text? or is it extracting the important frames out of a given video."
>
> In the interest of keeping the content within the page limit, we assumed knowledge of video summarization background on the part of the readers. With the extra page limit, we are now including the following context/background in the modified version:
>
> Videos have become an indispensable medium for capturing and conveying information in many sectors like entertainment (TV shows, movies, etc.), sports, personal events (birthday, wedding etc.), education (HOWTOs, tech talks etc.), to name a few. The increasing availability of cheaper and better video capturing and video storage devices have led to the unprecedented growth in the amount of video data available today. Most of this data, however, comes with a lot of redundancy, partly because of the inherent nature of videos (as a set of {\it many} images) and partly due to the 'capture-now-process-later' mentality. Consequently this has given rise to the need of automatic video summarization techniques which essentially aim at producing shorter videos without significantly compromising the quality and quantity of information contained in them. A Video Summarization technique aims to select important, diverse (non-redundant) and representative frames (called static video summarization) or shots (called dynamic video summarization) from a video to enable quicker and easier consumption of information contained in the video. For example, producing the highlights of a soccer video. In this work we focus on dynamic video summarization and introduce VISIOCITY, a new benchmarking dataset and evaluation framework as a step towards addressing some of the challenges in the literature.
>
> >> "From ln 199 to ln 204, the description of the preference of diversity is confusing. Why does one not want two similar-looking snippets to appear in a summary if they are indeed more important for video summary when connected? and why does one want them to be separated across time?"
>
> Given a fixed budget, a summary is supposed to be diverse and non redundant. A typical expectation is to not exhaust the budget by having too many similar snippets and hence diverse snippets are preferred. For example, say in a video of a TV show like ‘Friends’, you probably want your summary to contain different (diverse) snippets capturing different scenes/action and not want too many snippets conveying similar information in the limited budget. However, as explained in lines 204 to 209, in some other types of videos, say Surveillance, you may indeed indeed want even similar snippets to be in the summary, provided they correspond to different instants of time. The different proposed diversity measures capture these different semantics. In the modified version, we are adding some extra explanation in that section for better clarity.

---

> ### Author Response · Authors · 2021-07-11
> **Our responses (part 2)**
>
> [this is the continuation of our responses from part 1]
>
> >> "The paper lacks the visualizations of the concepts, tasks, procedures that are introduced in the paper, which otherwise could significantly help with the understanding of the authors' intent."
>
> Thanks for this suggestion. We are adding a schematic diagram right in the beginning to indicate the five key aspects of VISIOCITY:
> 1. that it is a large and diverse collection of videos across different categories, first of its kind;
> 2. that it contains more effective textual annotations as against frame/shot scores or ratings for each shot or ground truth summaries for each video;
> 3. that the textual annotations can be used in the proposed way to generate potentially many diverse ground truth summaries for training a supervised video summarization model;
> 4. that when a model learns from multiple losses (each measuring the deviation of a summary from a ground truth on some dimension), it performs better than the existing state-of-the-art approaches
> 5. that a video summary need not only be assessed against a ground truth summary, for there can be multiple ‘right answers’. A candidate summary doesn’t have to be disqualified as bad just because it wasn’t lucky enough to find a matching ground truth summary. Also, there can be different dimensions of assessing a video summary and hence for both reasons, a suite of evaluation measures computed using the textual annotation (indirect ground truth) is better than over dependence on a single measure (like F1), computed with respect to a ground truth summary;
>
> >> "In ln 186 to ln 192, the procedure for annotation is described. How were the precision and recall for the annotators determined? How were the ground truths annotated in the first place for analyzing the precision and recall? If there already existed some ground truths to determine these metrics, why is it necessary to hire annotators to re-annotate the data?"
>
> No ground truth existed prior to the annotation task. The annotators had to see the videos and mark the concepts applicable in every shot. In the verification round, a different set of annotators were tasked to go over the annotations to verify the correctness. Specifically, they were tasked to check if the marked concepts indeed exist in those shots in the video (and this is what we referred to here as precision - as in how accurate the annotations are?) and to check that important events in the video have been annotated (and this is what we referred to here as recall - as in have all events which are actually present in video been annotated?). We are adding this extra explanation in the modified version of the main text to avoid confusion.

---

> ### Author Response · Authors · 2021-07-14
> **Paper modified based on suggestions**
>
> As a continuation to our last two responses, we would like to inform that we have modified the main text in accordance to your valuable suggestions and our replies. Please let us know if there are any unaddressed concerns. Thanks for helping to make our work stronger.

---

### Official Review · Reviewer_cdfR · 2021-07-05

**Rating:** 6
**Confidence:** 4

**Strengths:**

- The new work contrasts with prior datasets for videos with diverse but very short videos on the one hand or longer videos of only a specific domain and lower diversity on the other.
- The paper accompanies the dataset with some analysis (although limited) and new evaluation metrics.
- The videos are associated with multiple types of annotations.
- Multiple models are compared over the new dataset, to give further information about where the open challenges are.

**Weaknesses:**

- The writing and presentation could be improved. More detail to back up the claims in the introduction could help making it more convincing.
- I think a comparative analysis of the dataset compared to other ones, in terms of providing different sorts of statistics, trends, and performance of similar models on different datasets would be really crucial to demonstrate the necessity of the dataset. Right now there is only written description in the related work section that briefly discusses how the new dataset is generally different from a list of other video datasets, but more extensive and especially quantitative comparison:
(1) multiple tables summarizing statistics for different datasets (beyond just number of videos and duration -- certainly a more careful analysis could allow for richer indications), (2) plots and visualizations showing the distribution within the dataset and partition to different domains etc. (3) a few examples of sample video frames from the data, to get a sense of their type, (4) more importantly, examples of specific annotations of the concepts as well as events for a specific video or frame; will all be very useful.

**Additional Feedback:**

Nope, all my feedback is above.

**Clarity:**

The content in the paper is relatively clear but I do feel writing and presentation quality could be improved. The writing right now is a bit more of like an outline without e.g. strong connections between paragraph or compelling presentation. I think more work on the writing style could strengthen the paper significantly, since the underlying dataset and content does definitely have the potential.
- The paper also mentioned the annotations are "indirect" but I'm not sure I understand why they are called that way. Is that because they don't give the explicit supervision of the summary, but rather are semantic annotation of the video's content? I don't know then if indirect is the best way to describe them.
- I'm not sure what exactly are mega-events and how they differ from standard events.

**Correctness:**

The introduction makes some claims without giving enough details to corroborate them. For instance, it says multiple times that the dataset is richly annotated, but without explaining specifically what annotations exist for it (during the introduction). Likewise, the explanation for why F1 metric isn't ideal is not detailed enough to convince a reader without a very close familiarity of the datasets and practices for video summarization. Perhaps this point is more about clarity than about correctness but I think it could be very useful to provide more context and explanation in the intro about the claims being made there.

**Documentation:**

Yes, there is a nice and easy-to-follow website that presents the dataset, gives statistics and visualizations, explains about the different annotations the dataset is accompanied with as well as the files structure, and highlights the advantages of the dataset.
In fact, I think the website does a much better job in presenting the dataset than the paper, which is actually great since it means the authors could easily fix the paper to match the website presentation.

**Ethics:**

There is the potential ethical concern of privacy -- in case personal information or videos of people being presented. This is especially relevant since the paper does mention featuring personal videos (of birthdays and weddings). Another important aspect is that of social diversity within the content (the paper does discuss visual technical diversity but of a different sort, e.g. of consecutive frames in a surveillance video). The paper unfortunately doesn't address this ethical aspects currently.

In addition, more information about how the videos have been collected would help clarifying some aspects of it (beyond saying that it contains "publicly available" videos) -- how the videos have been selected or filtered, was there any particular curation for annotations, etc.

**Relation To Prior Work:**

The paper does a good job in the related work section mentioning many related works and verbally comparing how the new dataset differs from each of them. It would be useful though to discuss it also in a higher-level perspective, presenting the general trends prior works have followed and what is new in the new work, rather than just comparing to a list of specific datasets one-by-one but without enough background and context.

The paper says that prior works use variety of evaluation metrics and then propose a new one. An additional discussion of the evaluation metrics proposed in the prior works and in which aspects they are different and similar to the new metrics will be very helpful.

**Summary And Contributions:**

The paper presents a new dataset called VISIOCITY for medium length video of varied domains. The key contributions are:
1. Introducing a new dataset for medium videos of different domains.
2. Exploring ways to augment the generate video summaries based on indirect annotations present within the dataset.
3.  Proposal a new evaluation scheme that is claimed to be closer to human judgement, by studying desirable properties of summaries.

In terms of rating I overall feel the paper is marginally above acceptance: the dataset could be helpful for the community and advance this research area, and it comes with nice analysis and evaluation framework, yet I feel both the paper and the research of the dataset could be polished and improved further to become clearer and more solid.

Update: I thoroughly read the author's response as well as the other reviews, and considering all these I overall would like to keep my current score (considering both the improvements on the one hand made by the authors and the weaknesses discussed by the other reviewers)

---

> ### Author Response · Authors · 2021-07-11
> **Our response to questions/concerns raised (part 1)**
>
> We thank you for your appreciation of VISIOCITY and for your valuable inputs. We address your questions/concerns below. We are incorporating the changes in the modified version and shall submit the same before the discussion phase ends.
>
> >> “The writing and presentation could be improved".....”In fact, I think the website does a much better job in presenting the dataset than the paper, which is actually great since it means the authors could easily fix the paper to match the website presentation.”
>
> Thanks for the appreciation. We understand the concern on clarity. Thanks to the extra one page limit, we are now enhancing the clarity of our presentation in the following ways:
> 1. We are adding a schematic diagram right in the beginning to indicate the five key aspects of VISIOCITY:
> - that it is a large and diverse collection of videos across different categories, first of its kind;
> - that it contains more effective textual annotations as against frame/shot scores or ratings for each shot or ground truth summaries for each video;
> - that the textual annotations can be used in the proposed way to generate potentially many diverse ground truth summaries for training a supervised video summarization model;
> - that when a model learns from multiple losses (each measuring the deviation of a summary from a ground truth on some dimension), it performs better than the existing state-of-the-art approaches
> - that a video summary need not only be assessed against a ground truth summary, for there can be multiple ‘right answers’. A candidate summary doesn’t have to be disqualified as bad just because it wasn’t lucky enough to find a matching ground truth summary. Also, there can be different dimensions of assessing a video summary and hence for both reasons, a suite of evaluation measures computed using the textual annotation (indirect ground truth) is better than over dependence on a single measure (like F1), computed with respect to a ground truth summary;
> 2. To keep the content under the page limit, we had to deliberately move some content from the main paper to the dataset website (https://visiocity.github.io/). We are ensuring that the main paper now contains enough information for clarity.
> 3. Towards the end of the “Introduction and Motivation” we are adding a paragraph describing the outline of the paper and connections between the sections. We are also renaming some paragraph / section titles for clarity of flow.
>
> >> “performance of similar models on different datasets”
>
> Thanks for the suggestion. We are adding a table in the modified version giving results reported by vsLSTM, VASNET and DR-DSN on SUMME and TVSUM as follows:
> \begin{tabular}{ccc}
> & SUMME & TVSUM \\
> vsLSTM & 41.6 & 57.9 \\
> VASNET & 51.09 & 62.37 \\
> DR-DSN & 43.9 & 59.8 \\
> \end{tabular}
> These methods report F1 scores on TVSUM and SUMME and hence are not directly comparable in our settings. We see that while they measured their success on SUMME and TVSUM, their strengths and weaknesses are better highlighted when tested on VISIOCITY. This is due to two reasons: 1) SUMME and TVSUM are tiny datasets while VISIOCITY is huge compared to them and, 2) we evaluate using a richer evaluation framework.
>
> >> “(1) multiple tables summarizing statistics for different datasets (beyond just number of videos and duration -- certainly a more careful analysis could allow for richer indications)"
>
> This is a very good suggestion. We are enhancing Tab. 1 by adding following columns:
> - Number of categories,
> - Nature of annotations (rating//ground truth summary)
> - Unit of annotation (frame/shot/snippet),
> - Number of ground truth summaries provided
>
> >> “(2) plots and visualizations showing the distribution within the dataset and partition to different domains etc.”
>
> Thanks for this suggestion. We are enhancing Tab. 2 by adding a column indicating the types and number of videos within each category of VISIOCITY.
>
> >> “(3) a few examples of sample video frames from the data, to get a sense of their type,”
>
> Sure. This is already on the website, but we are adding it in the modified version of the main text as well. Thanks for pointing it out.
>
> >> “(4) more importantly, examples of specific annotations of the concepts as well as events for a specific video or frame; will all be very useful.”
>
> This will indeed be good for readers to get a better sense of the annotations. We had space concerns, but given the extra page, we are now adding it to the main text.
>
> >>“I'm not sure what exactly are mega-events and how they differ from standard events.”
>
> They are indeed standard events. As discussed in L. 147-153, the notion of mega-events allows for marking consecutive snippets as being a part of an event and model continuity which is essential for visually pleasing summaries. The prefix ‘mega’ refers to the fact that it is not an annotation of a snippet per se but is a higher level annotation corresponding to a group of snippets.

---

> > ### Comment · Reviewer_cdfR · 2021-07-11
> > **Thanks for the reply**
> >
> > Thank you for following up on my comments and working on incorporating them and improving the paper. I'm looking forward to the new version of the paper!

---

> ### Author Response · Authors · 2021-07-11
> **Our response (part 2)**
>
> [this is a continuation from previous comment]
>
> >> “The introduction makes some claims ... it says multiple times that the dataset is richly annotated...the explanation for why F1 metric isn't ideal is not detailed enough...it could be very useful to provide more context and explanation in the intro about the claims being made there.”
>
> Thanks for these suggestions. While we have elaborated on certain details later in the paper or on the Website, we agree that some of these can be explained earlier on. We are enhancing the Introduction with this in mind and also via a schematic diagram as mentioned above.
>
> >> "The paper also mentioned the annotations are "indirect" but I'm not sure I understand why they are called that way."
>
> Yes. As mentioned in L.136-137 of the main text, “the ground truth in VISIOCITY is not direct in form of the user summaries, but indirect in form of concepts marked for each snippet”. In a supervised setting, the video and its summary form the training examples. The ground truth thus expected by such models is the ground truth summaries. Datasets like You Tube 1 [2], You Tube 2 [2], SUMME [6] do provide such direct ground truth. However, as discussed in  L.136-141, indirect ground truth in the form of concept annotations and NOT as summaries themselves, offers several advantages. This in fact was one of the recommendations by [27], specifically in their Sec. 5.4.2.
>
> >> "The paper does a good job in the related work section mentioning many related works and verbally comparing how the new dataset differs from each of them. It would be useful though to discuss it also in a higher-level perspective, presenting the general trends prior works have followed and what is new in the new work, rather than just comparing to a list of specific datasets one-by-one but without enough background and context."
>
> Very valid point indeed. However, as far as the related work specifically on datasets is concerned, unfortunately it is more about the characteristics of the datasets themselves than any trend per se and hence it had to be listed that way. One of the prominent problems in Video Summarization literature has been a lack of a standardized benchmarking dataset. Because of this, in proposing new techniques of summarization, researchers often created new datasets, not necessarily “evolving” the datasets in any specific ways. Hence while it is possible to talk about trends in techniques, it is difficult to talk about any “trends” in the datasets per se beyond their characteristics and limitations.
>
> >> "The paper says that prior works use variety of evaluation metrics and then propose a new one. An additional discussion of the evaluation metrics proposed in the prior works and in which aspects they are different and similar to the new metrics will be very helpful."
>
> Thanks for the suggestion. We agree that including this will be helpful to appreciate the current contribution. We are adding the following in the modified version of the main text: “Early approaches like Ma et al. “A user attention model for video summarization”, ACM Multimedia 2002, involved user studies but suffered the obvious demerit of cost and reproducibility. With a move to automatic evaluation, every new technique of video summarization came with its own evaluation criteria making it difficult to compare results across different techniques. VIPER (D. Doermann and D. Mihalcik. “Tools and techniques for video performance evaluation”, ICPR 2000), addressed the problem by defining a specific ground truth format which makes it easy to evaluate a candidate summary, while SUPERSEIV (Huang et. al. “Automatic  performance  evaluation  for video  summarization”, Technical  report,  Inst. for Advanced Computer Studies, 2004) was an unsupervised technique to evaluate video summarization algorithms that perform frame ranking. VERT (Li et al, “Vert:  automatic evaluation of video summaries”, ACM Multimedia 2012) on the other hand was inspired by BLEU in machine translation and ROUGE in text summarization. Other techniques include pixel-level distance between keyframes (Khosla et. al. “Large-scale video summarization using web-image  priors”, CVPR 2013), objects of interest as an indicator of similarity (Lee et. al., “Discovering important people and objects for egocentric video summarization”, CVPR 2012) and precision-recall scores over key-frames selected by human annotators (Gong et. al., “Diverse  sequential  subset  selection  for  supervised video summarization”, NIPS 2014).

---

> ### Author Response · Authors · 2021-07-14
> **Paper modified based on suggestions**
>
> As a continuation to our last two responses, we would like to inform that we have modified the main text in accordance to your valuable suggestions and our replies. Please let us know if there are any unaddressed concerns. Thanks for helping to make our work stronger.

---

### Official Review · Reviewer_agV6 · 2021-07-07
**Larger dataset for a better video summarization and more comprehensive evaluation metrics**

**Rating:** 5
**Confidence:** 2

**Strengths:**

- Their dataset is larger and more diverse than other existing public video summarization datasets when considering the several dimensions altogether: number of videos, average duration of videos, and number of categories. It should enable researchers in the field to develop more complex models to tackle the problem that would otherwise overfit due to the lack of data. The data is publicly available (upon request).
- They tackle the difficulty of collecting user summaries of videos by automatically generating a reference groundtruth for each video from indirect annotations of concepts in each video. Having one reference groundtruth instead of the one of multiple users, eases the training of supervised models for video summarization. The results are also more interpretable thanks to the proposed measures (diversity, mega-event continuity, and importance) that serve as both loss terms to the supervised learning and as final evaluation metrics.



**Weaknesses:**

Not any clear weaknesses in the dataset or the evaluation framework. The main weakness is the clarity of writing, which is not enough for me to fully understand some parts related to the baseline and evaluation.

**Additional Feedback:**

None

**Clarity:**

- It is sometimes hard to follow. The itemization of paragraphs into sections is not very appealing and lacks some paper content outlining both at the end of the "Introduction & Motivation" section and at the beginning of each section to know exactly what to expect in the following text.
- The mathematical notation is heavily based on the one from the following work:
"Gygli, Michael, Helmut Grabner, and Luc Van Gool. "Video summarization by learning submodular mixtures of objectives." Proceedings of the IEEE conference on computer vision and pattern recognition. 2015."
But is not as polished and some details are missing so it difficults the reading. For instance, I could not find how $d_{ij}$ is calculated or $sim$ function is defined. More importantly, it is still not clear to me which if the proposed measures (Table 3) are used in the optimization as loss terms or just as evaluation metrics.

**Correctness:**

Everything looks correct: the construction of videos (despite no details being provided), the evaluation of the methods, and the experiments are illustrating the correctness of the groundtruth generation and the performance of the baseline VISIOCITY-SUM in the VISIOCITY dataset.

**Documentation:**

- Is there sufficient detail on data collection and organization, availability and maintenance, and ethical and responsible use? Little to no info about collection and organization of data in the main document beyond if the data is public or from their own. In the checklist and in their webpage, their state the publicly available videos have been collected from YouTube. Not information about data characteristics: whether if they reduce spatial resolution, speed, or something. It is not 100% clear in the webpage or in the paper if they provide the videos or only their YouTube links. So not sure about their future availability or maintenance. For their privately collected data, i.e. surveillance category, people are identifiable but the videos are only provided upon request.
- They provide an URL but not for reviewers, but for the general public to access the data. As a reviewer I cannot access to it without requesting access.
- They provide code to reproduce the experiments.



**Ethics:**

No.

**Relation To Prior Work:**

Their baseline method is basically based on "Gygli, Michael, Helmut Grabner, and Luc Van Gool. "Video summarization by learning submodular mixtures of objectives." Proceedings of the IEEE conference on computer vision and pattern recognition. 2015.". The contributions in the baseline -- if any -- with respect to that work are not clear.

**Summary And Contributions:**

This paper introduces a new benchmarking dataset VISIOCITY and several evaluation criteria to favourite better, more explainable and comprehensive summarization of videos. Compared to other video summarization datasets, the authors do not offer the largest number of videos, average duration, or number of categories, but an overall compromise among all those characteristics that make the dataset quite appealing (67 videos, 55 minutes of average duration, and 6 categories). Videos were mostly collected from public sources, but also their own for the surveillance category. The annotations are not provided in the form of user summaries, but as high-level semantic textual concepts. These concepts enable automatic summary groundtruth generation that can be different depending on the mixture weights of criteria (different kinds of diversity, mega-event continuity, or importance/interestingness). These are same criteria used for the evaluation of summarization models, complementing the very common F1 metric. Their baseline implementation for video summarization using their data and automatic groundtruth show the reliability of their approach surpassing other state-of-the-art works when applied to their dataset.

---

> ### Author Response · Authors · 2021-07-11
> **Our responses to questions/concerns raised**
>
> Thank you for your valuable comments and for appreciating our contribution. We address your questions/concerns below. We are incorporating these changes in the modified version and shall submit the same before the discussion phase ends.
>
> >> “Their baseline method is basically based on "Gygli, Michael, .. computer vision and pattern recognition. 2015.". The contributions in the baseline -- if any -- with respect to that work are not clear.”
>
> We would like to clarify that we do not use Gygli et. al., "Video summarization by learning submodular mixtures of objectives.", CVPR 2015 as a baseline per se, in the sense of comparing against it or trying to perform better than it. Rather, we employ the strategy of large margin learning of mixtures as proposed in that work and earlier works such as (Lin & Bilmes UAI 2012, Tschiatchek et. al. NIPS 2014) and apply it with a loss defined as a linear combination of the proposed measures. We show that even a simple recipe like this produces better summaries than some of the existing state of the art techniques when tested using the proposed evaluation measures on the challenging VISIOCITY dataset. We are adding this clarification in the modified version.
>
> >> "I could not find how dij  is calculated or sim function is defined. More importantly, it is still not clear to me which if the proposed measures (Table 3) are used in the optimization as loss terms or just as evaluation metrics.”
>
> In L. 210, $d_ij$ is a measure of dissimilarity between snippets $i$ and $j$ which are represented as concept vectors (a binary vector with a 1 at the indices corresponding to the concepts contained in the snippet). Intersection over union (IOU) between these two vectors gives the similarity and one minus that gives us the measure of dissimilarity. We are adding this clarification in the modified version.
>
> The $sim()$ in the facility-location function mentioned at L.305 refers to the cosine similarity between two snippets represented as concept vectors. During training and inference, these concept vectors are computed based on the detections from a YOLO object detection model. We regret missing this important detail in the paper and are adding it in the modified version of the paper.
>
> As far as measures defined in Tab. 3 are concerned, we use them for three purposes:
> - To compute the ground truth summaries (as described currently in Sec. 5)
> - To compute the loss in the optimization (as mentioned in L.315-317) [we use all the measures]
> - As evaluation metrics to compute the scores of any summary (example as reported in Tab. 4 and Tab. 5)
>
> >> “The itemization of paragraphs into sections is not very appealing and lacks some paper content outlining both at the end of the "Introduction & Motivation" section and at the beginning of each section to know exactly what to expect in the following text.”
>
> Thanks to the extra one page limit, we are now enhancing the clarity of our presentation in the following ways:
> 1. Towards the end of the “Introduction and Motivation” we are adding a paragraph describing the outline of the paper and connections between the sections. We are also renaming some paragraph / section titles for clarity of flow.
> 2. We are adding a schematic diagram right in the beginning to indicate the \textbf{five key aspects of VISIOCITY}:
> - that it is a large and diverse collection of videos across different categories, first of its kind;
> - that it contains more effective textual annotations as against frame/shot scores or ratings for each shot or ground truth summaries for each video;
> - that the textual annotations can be used in the proposed way to generate potentially many diverse ground truth summaries for training a supervised video summarization model;
> - that when a model learns from multiple losses (each measuring the deviation of a summary from a ground truth on some dimension), it performs better than the existing state-of-the-art approaches
> - that a video summary need not only be assessed against a ground truth summary, for there can be multiple ‘right answers’. A candidate summary doesn’t have to be disqualified as bad just because it wasn’t lucky enough to find a matching ground truth summary. Also, there can be different dimensions of assessing a video summary and hence for both reasons, a suite of evaluation measures computed using the textual annotation (indirect ground truth) is better than over dependence on a single measure (like F1), computed with respect to a ground truth summary;
> 3. To keep the content under the page limit, we had to deliberately move some content from the main paper to the dataset website (https://visiocity.github.io/). We are ensuring that the main paper now contains enough information for clarity.

---

> ### Author Response · Authors · 2021-07-11
> **Our response (part 2)**
>
> [this is a continuation of the previous response]
>
> >> “Little to no info about collection and organization of data in the main document ......Not information about data characteristics: whether if they reduce spatial resolution, speed, or something. It is not 100% clear in the webpage or in the paper if they provide the videos or only their YouTube links. So not sure about their future availability or maintenance.....”
>
> The videos (available at https://tinyurl.com/visiocity-videos) and also through the project website (https://visiocity.github.io/) are organized in folders, each folder corresponding to a category. The videos are available as original mp4/avi files as downloaded/collected. We do not post-process on top of them.
>
> We provide the videos and not just the URLs. The videos are available in Google Drive and shall always be available and maintained.
>
> We are adding this information explicitly in the modified version of the main text.
>
> >> “They provide an URL but not for reviewers, but for the general public to access the data. As a reviewer I cannot access to it without requesting access.”
>
> We would like to clarify that the Google Form on the web page to access/download the dataset is just to collect some information about those interested in the dataset. As a reviewer you can always fill in say, “NeurIPS Reviewer” there and some dummy values of Affiliation, Email etc. and as soon as you submit the form, you get the links to access the dataset. Or you may also access them here:
>
> - All videos of VISIOCITY: https://tinyurl.com/visiocity-videos
> - JSON files of all annotations for all videos and all categories: https://tinyurl.com/visiocity-annotations
> - Human Summaries for preliminary analysis: https://tinyurl.com/visiocity-human

---

> ### Author Response · Authors · 2021-07-14
> **Paper modified based on suggestions**
>
> As a continuation to our last two responses, we would like to inform that we have modified the main text in accordance to your valuable suggestions and our replies. Let us know if there are any unaddressed concerns. Thanks for helping to make our work stronger.

---

### Decision · Program_Chairs · 2021-07-26

**Decision:**

Reject

**Comment:**

The reviewers share critiques regarding the writing, presentation and analyses. On the other hand all reviewers agree on the high quality and unique value of the proposed dataset for video summarization compared to previous alternatives. The authors did a great job discussing with reviewers regarding their complains. However the paper still needs some improvement to be ready for publication. Unfortunatelly it can not be accepted in its current for for round 1. We encourage authors to carefully improve current version of the paper and submit it to round 2.